# Enhancing Out-of-Distribution Detection Using Synthesized OoD Samples from In-Distribution Training Data

## Abstract

Conventional out-of-distribution (OOD) detection methods typically rely on identifying deviations from in-distribution (InD) patterns, often necessitating complex density estimation and threshold tuning. These approaches are inherently difficult to validate and brittle across different deployment scenarios. Motivated by the observation that OOD embeddings geometrically collapse toward the origin while ID embeddings form orthogonal clusters, we propose a novel framework that synthesizes auxiliary OOD embeddings exploiting this structural difference. By augmenting the training process with these OOD synthetic representations, we reformulate OOD detection as a direct supervised binary classification task (i.e., OOD vs ID). Empirical evaluations demonstrate that this approach significantly enhances detection robustness, achieving an average performance improvement compared to state-of-the-art baselines.

## 1 Introduction

Modern deep learning (DL) classifier models generalize well when test inputs share the same distribution of discriminative features as the training data. However, performance degrades sharply on out-of-distribution (OoD) inputs—cases where the class-discriminative features differ from those seen during training—which are common in real-world settings. Moreover, in-distribution (InD) training data consist of a closed set of classes (i.e., a fixed number of classes derived from annotations and data aggregation), whereas OoD data represent an open, unavailable set of classes. Current approaches focus on accurately identifying InD data and categorizing all other data as OoD by assuming a clear separation between InD and OoD embeddings (see fig. 1a). These models typically employ scoring methods that assign high likelihoods to areas densely populated with InD embeddings and default low likelihoods elsewhere, aiming to detect OoD data when the likelihood score falls below a predefined threshold. Validating these scoring methods and determining the appropriate threshold requires OoD data. However, because the OoD space is inherently open-set, constructing a representative OoD benchmark for various architectures and training distributions is nontrivial. Consequently, an inadequate OoD dataset may result in suboptimal validation of scoring methods and thresholds, potentially diminishing the effectiveness of OoD detection. To address these challenges, we propose generating auxiliary OoD embeddings directly from InD data and using them to design, train, and validate a binary classifier that separates InD from OoD. This yields a more accurate and validated parameterization of the decision boundary between the two. To generate OoD embeddings using only InD data, we leverage a key geometric property of neural network representations in the logit space Komini & Girdzijauskas (2024). Specifically, the embeddings corresponding to the InD samples tend to align along distinct, approximately orthogonal axes, forming well-separated class-wise clusters that push toward high positive activation values for their respective class dimensions (see fig. 1a). In contrast, OoD embeddings, which lack strong affinity to any learned class-specific features, collapse toward the center of the logit embedding space, exhibiting low-magnitude and less directional activation patterns (see fig. 1a). During training, the classifier learns to generalize the optimal class-specific discriminative features from InD samples to maximize the logit corresponding to their ground-truth class, implicitly driving them away from the origin and reducing cross-class interference. In contrast, OoD inputs do not exhibit any of the learned class-specific features, causing the classifier to default to low-confidence predictions that lie near the center of the logit embedding

space Komini & Girdzijauskas (2024). To leverage this geometric separation between InD and OoD embeddings, we employ a leave-one-class-out training paradigm. In this setup, we remove a single class from the training set and train the classifier solely on the remaining classes (see fig. 2). Because the model is never exposed to the held-out class during training, inputs from this class do not activate any of the learned discriminative features associated with the InD classes. As a result, their embeddings naturally map toward the central region of the logit space, closely mimicking the behavior of true OoD samples. This held-out class, therefore, functions as a principled proxy for OoD data, allowing us to generate realistic OoD-like embeddings without requiring access to any external OoD dataset (see fig. 1b). We then reinitialize the model and retrain it on the full training set to produce InD embeddings. By synthesizing auxiliary OoD embeddings from InD data, we train a simple binary discriminator that explicitly separates InD from OoD representations (see fig. 1b). This reframes the task from modeling the complex InD embedding manifold to estimating a direct decision boundary between InD and OoD. In addition, the generated OoD surrogates enable stronger validation, yielding more reliable OoD detection at deployment compared to prior methods. We demonstrate improved performance across multiple benchmarks and dataset types. Our method also maintains consistent performance across diverse model architectures. Our contributions are twofold:
1. We introduce a framework that synthesizes surrogate OOD embeddings directly from InD data, supported by theoretical analysis.
2. We demonstrate that a binary classifier trained on these auxiliary embeddings effectively generalizes to real-world OOD data, eliminating the need for labeled outliers.

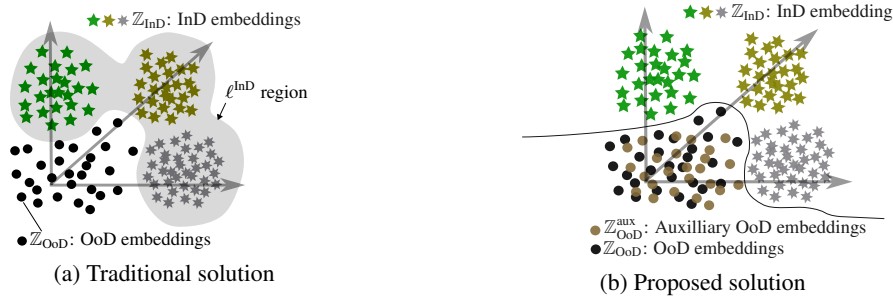

(a) Traditional solution
(b) Proposed solution

Figure 1: **OoD detection.** Traditional methods assign an InD likelihood score $\ell^{\text{InD}}$ to embeddings (see fig. 1a) and apply a threshold to separate InD (i.e., $\mathbb{Z}_{\text{InD}}$) from OoD (i.e., $\mathbb{Z}_{\text{OoD}}$). In contrast, we generate auxiliary OoD logit embeddings (i.e., $\mathbb{Z}^{\text{aux}}_{\text{OoD}}$) that approximate the true OoD regions and learn an explicit boundary between InD and OoD via a parametric function $g_\eta : \mathbb{R}^K \to \mathbb{R}^2$ (see fig. 1b).

## 2 GENERATION OF AUXILIARY OOD LOGITS

The region of OoD logit embeddings (i.e., $\mathbb{Z}_{\text{OoD}}$) typically lies near the center of the logit space Komini & Girdzijauskas (2024), but parameterizing this region accurately requires access to OoD data, which is inherently unavailable during training. To address this, we propose generating auxiliary OoD logits (i.e., $\mathbb{Z}^{aux}_{\text{OoD}}$) using InD data. Our approach leverages an auxiliary discriminative model $f^{\text{Aux}}_\theta : \mathcal{X} \to \mathbb{R}^{K-1}$, trained on a subset of InD classes $\mathcal{D}^{\text{aux}}_{\text{train}} = \cup_{k=1}^{K-1} \mathcal{C}_k$, while deliberately excluding one class (i.e., $\mathcal{C}_K$) (see fig. 2a and Appendix A). By design, the auxiliary model (i.e., $f^{\text{Aux}}_\theta$) learns only the discriminative features of the included classes, rendering the excluded class's samples $\mathcal{D}^{\text{aux}}_{\text{OoD}} = \mathcal{D}_{\text{InD}} \setminus \mathcal{D}^{\text{aux}}_{\text{train}}$ effectively OoD with respect to $f^{\text{Aux}}_\theta$. The decision to exclude a single class ensures that the excluded samples serve as near-OoD data, retaining non-discriminative features while lacking class-specific discriminative attributes generalized by $f^{\text{Aux}}_\theta$. This minimizes the risk of overlap with InD logit embeddings and reduces potential discrepancies in logit displacement between models trained on subsets versus the full dataset. Considering that the auxiliary model (i.e. $f^{\text{Aux}}_\theta$) is trained in almost the same classes as the main classifier (i.e. $f_\theta$), with the exception of just one class, it provides a reliable approximation of the model trained in the entire classes. Furthermore, under the class-conditional independence assumption (i.e., $\mathcal{C}_i \perp\!\!\!\perp \mathcal{C}_j, \forall i \neq j$ see Assumption 1 in Appendix F), we can derive an upper bound on the approximation error (see Lemma 1 in Appendix F) that is minimal when a single class is excluded. The resulting OoD logits, $\mathbb{Z}^{\text{aux}}_{\text{OoD}} = \{f^{\text{aux}}_\theta(x) \mid x \in \mathcal{C}_K\}$,

provide a principled approximation of the OoD region in the reduced $\mathbb{R}^{K-1}$ logit space embedding. Furthermore, due to class balance and the saturation of loss across all in-distribution (InD) classes in both the primary and auxiliary models, the resulting logits are evenly displaced from the center of the feature space Dang et al. (2024) (see fig. 3 and Appendix I for an extended empirical analysis). Consequently, the OoD logits are also expected to exhibit a uniform displacement across classes, which we validate empirically in fig. 3 and Appendix I where the same behavior extends to auxiliary OoD data. We therefore approximate the missing dimension by sampling with replacement from the available logit embeddings (see Algorithm 1 in Appendix D). To obtain InD logits, we train a model $f_\theta : \mathcal{X} \to \mathbb{R}^K$ with the same architecture as the auxiliary model, using the full training dataset $\mathcal{D}_{\text{InD}} = \cup_{k=1}^K \mathcal{C}_k$ (see fig. 2b). The trained model's outputs on $\mathcal{D}_{\text{train}}$ then serve as the InD logit embeddings $\mathbb{Z}_{\text{InD}} = \{ f_\theta(x) \mid x \in \mathcal{D}_{\text{InD}} \}$. We can now train a binary classifier that can tell the auxiliary OOD apart from the InD logit embedding (i.e., $\mathbb{Z}_{\text{OoD}}^{\text{aux}}$ vs $\mathbb{Z}_{\text{InD}}$) as in Appendix E.

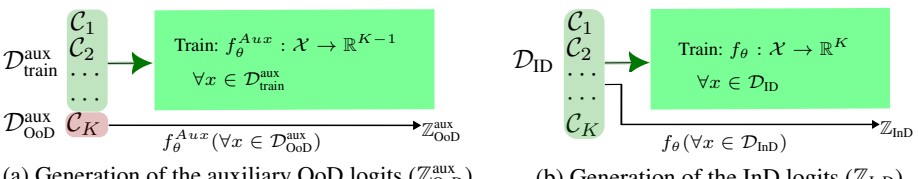

(a) Generation of the auxiliary OoD logits ($\mathbb{Z}_{\text{OoD}}^{\text{aux}}$).     (b) Generation of the InD logits ($\mathbb{Z}_{\text{InD}}$).

Figure 2: **Generation of auxiliary OoD and InD embeddings:** The auxiliary model $f_\theta^{\text{aux}} : \mathcal{X} \to \mathbb{R}^{K-1}$ (see fig. 2a) is trained on the reduced dataset $\mathcal{D}_{\text{train}}^{\text{aux}} = \bigcup_{k=1}^{K-1} \mathcal{C}_k$, explicitly excluding class $\mathcal{C}_K$. To approximate the OoD embeddings logits $\mathbb{Z}_{\text{OoD}}^{\text{aux}}$, samples from the held-out class $\mathcal{C}_K$ are passed through $f_\theta^{\text{aux}}$. The main model $f_\theta : \mathcal{X} \to \mathbb{R}^K$ (see fig. 2b) is then trained on the full InD dataset $\mathcal{D}_{\text{InD}} = \bigcup_{k=1}^K \mathcal{C}_k$ to produce InD embeddings logits $\mathbb{Z}_{\text{InD}}$.

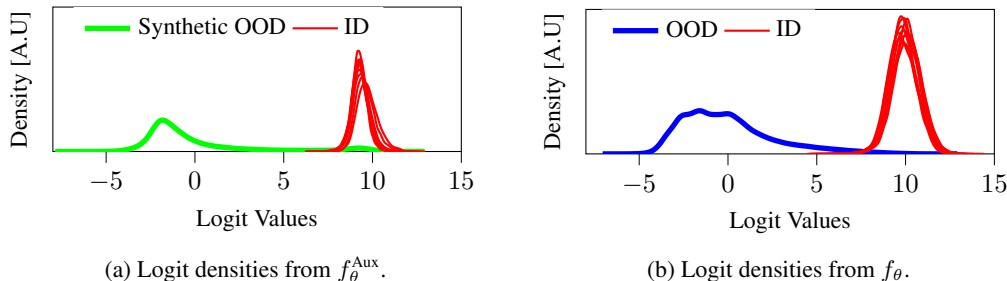

(a) Logit densities from $f_\theta^{\text{Aux}}$.     (b) Logit densities from $f_\theta$.

Figure 3: Logit density plots produced by the auxiliary classifier $f_\theta^{\text{Aux}}$ (trained on $K-1$ CIFAR-10 classes) and the full classifier $f_\theta$ (trained on all $K$ classes). Figure 3a presents InD logit distributions for CIFAR-10 classes 1–9 and synthetic OoD data (class 10), while fig. 3b displays distributions for all 10 classes as InD and SVHN as OoD. All results shown here use ResNet-34 as the backbone.

Unlike previous methods, our approach uniquely exploits the auxiliary model's discriminative feature learning in a straightforward, controlled manner, enabling reliable validation of InD/OoD separation without external data. This method circumvents the need for explicit OoD samples and ensures that the generated OoD logits align with the model's inherent generalization behavior. To approximate the OoD distribution, we exclude one class from training and use its samples as auxiliary OoD data. Assuming that the training classes are class balanced are pairwise contextually distinct (class-conditional independence assumption, i.e. $\mathcal{C}_i \perp\!\!\!\perp \mathcal{C}_j \ \forall i \neq j$ see Assumption 1 in Appendix F), the specific class excluded is inconsequential: any choice yields approximately the same auxiliary OoD logits (see fig. 5 ). Furthermore, because the auxiliary OoD logit embeddings are expected to be (approximately) linearly separable from each class-wise cluster of InD logit embeddings (in $\mathbb{R}^K$), we can validate their quality by quantifying this separation—for example, using density plots as in fig. 3 and Appendix I.

## 3 EXPERIMENTS

Our experiments address two questions: (i) how the proposed method compares with established alternatives, and (ii) whether its performance is sustained across different model families. To assess the effectiveness of our scoring rule, we conduct a head-to-head comparison against representative OoD detectors that operate on either logits or pre-logit embeddings (see Appendix C for an extended explanation of each baseline method). Among the baselines, MSP Hendrycks & Gimpel (2017), Mahalanobis Lee et al. (2018), and KNN Sun et al. (2022) were implemented by us, while the remaining methods were obtained from public repositories [1]. Importantly, unlike most baselines, our method requires only InD data for training and validation, without any access to external OoD datasets. Unless stated otherwise, we train a ResNet-18 He et al. (2016) on CIFAR-10 Krizhevsky et al. as the in-distribution (InD) dataset, and evaluate out-of-distribution (OoD) detection on SVHN Netzer et al. (2011), iSUN Xu et al. (2015), LSUN Yu et al. (2016), Textures Cimpoi et al. (2014), and Places365 Zhou et al. (2018) (see table 1). Existing methods typically produce only an InD score, denoted $\ell^{\mathrm{InD}}(z \mid \eta)$, whereas our approach also defines an OoD score $\ell^{\mathrm{OoD}}(z \mid \eta)$. For a fair comparison, we follow common practice and report FPR@TPR95 (false positive rate at 95% true positive rate) and AUROC computed using $\ell^{\mathrm{InD}}(z \mid \eta)$. In addition, we train a lightweight discriminator in InD and auxiliary-OoD embeddings: a two-hidden-layer MLP (i.e., $g_\eta : \mathbb{R}^K \to \mathbb{R}^2$) trained with cross-entropy for binary InD/OoD classification as detailed in Algorithm 2.

Table 1: AUROC and FPR over different methods (**ResNet18** trained on **CIFAR10**).

| method | SVHN | | LSUN | | iSUN | | Textures | | Places365 | | AVERAGE | |
|---|---|---|---|---|---|---|---|---|---|---|---|---|
| | FPR↓ | AUROC↑ | FPR↓ | AUROC↑ | FPR↓ | AUROC↑ | FPR↓ | AUROC↑ | FPR↓ | AUROC↑ | FPR↓ | AUROC↑ |
| MSP Hendrycks & Gimpel (2017) | 59.66 | 91.25 | 45.21 | 93.80 | 54.57 | 92.12 | 66.45 | 88.50 | 62.46 | 88.64 | 57.67 | 90.86 |
| ODIN Liang et al. (2018) | 20.93 | 95.55 | 7.26 | 98.53 | 9.84 | 96.54 | 56.40 | 86.21 | 63.04 | 86.57 | 36.16 | 92.30 |
| Energy Liu et al. (2020) | 54.41 | 91.22 | 10.19 | 98.05 | 27.52 | 95.59 | 55.23 | 89.37 | 42.77 | 91.02 | 38.02 | 93.05 |
| GODIN Hsu et al. (2020) | 15.51 | 96.01 | 9.46 | 97.91 | 13.20 | 94.69 | 40.61 | 89.68 | 32.80 | 93.52 | 22.80 | 93.52 |
| Mahala Lee et al. (2018) | 9.24 | 97.80 | 67.73 | 73.61 | 62.03 | 46.98 | 23.21 | 91.21 | 69.56 | **97.34** | 46.34 | 86.50 |
| KNN Sun et al. (2022) | 24.53 | 95.96 | 25.29 | 95.69 | 25.55 | 95.26 | 27.57 | 94.71 | 50.90 | 89.14 | 30.77 | 94.15 |
| Ours | **8.19** | **98.79** | **8.11** | **98.90** | **8.20** | **98.99** | **22.24** | **95.13** | **30.73** | 96.32 | **15.494** | **97.62** |

Furthermore, we extended our evaluation to various architectural backbones, including multiple versions of ResNet and DenseNet (see Appendix I). Additionally, we analyzed the impact of the specific classes excluded during training, which serve as auxiliary OOD samples (see Appendix G). We also conducted a comprehensive ablation study to assess the sensitivity of our method to the number of InD training classes (see Appendix H). Finally, it is crucial to emphasize that robust OOD detection depends on the classifier's ability to generalize class-specific features; this capability is currently empirically verified by validation accuracy on the InD test set.

## 4 CONCLUSION

We introduced a novel framework for OoD detection that learns an explicit InD-OoD boundary in logit space embeddings. In contrast to methods that rely on surrogate scores and ad-hoc threshold tuning, our detector exploits the geometry of logit space: InD logits form well-separated, class-specific clusters at approximately equal distances from the origin, while OoD logits tend to concentrate near the center. We further showed that DL models can approximate OoD behavior by training on a single held-out class and reusing its samples as *auxiliary OoD* data. Under the assumption that class-specific discriminative features in the InD data are statistically independent, we showed analytically and empirically that excluding fewer classes yields increasingly faithful approximations of the fully trained model. Building on these insights, we proposed a simple strategy to generate auxiliary OoD embeddings that closely track the true OoD region. The method is *model-agnostic*—performing consistently across architectures—and *data-agnostic*—exhibiting the expected trends on diverse datasets—while reducing dependence on heuristic post-processing. This makes it particularly appealing for deployment in realistic settings where OoD distributions cannot be anticipated in advance. Existing OoD methods have not explicitly targeted enlargement of the InD–OoD separation in representation space. Some encourage separation indirectly via supervised contrastive training Sun et al. (2022) or feature normalization Müller & Hein (2025), but these remain heuristic.

---

[1]https://github.com/kkirchheim/pytorch-ood?tab=readme-ov-file

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

APPENDIX

## A   Background

In practice, it is not feasible to parameterize the distribution of class-specific discriminative features of InD data (i.e., $\mathcal{X} \times \mathcal{Y}$) as they reside onto an unknown manifold; instead, we have an empirical representation through annotated training data $\mathcal{D}_{\text{InD}} = \{(x_i, y_i)\}_{i=1}^{N}$. Under the manifold assumption, we assume that InD data form class-specific regions $\mathcal{D}_{\text{InD}} = \cup_{k=1}^{K} \mathcal{C}_k$ in the feature space $\mathcal{X}$ (where $K$ is the total number of classes), structured according to discriminative features unique to each class Roweis & Saul (2000); Tenenbaum et al. (2000); Ma & Fu (2011); Weng (2021). Although the precise structure of this feature space $\mathcal{X}$ is not fully known, the annotated training set $\mathcal{D}_{\text{InD}}$ serves as a substitute to learn its topology.

In practice, we train a parametric DL model $f_\theta : \mathcal{X} \to \mathbb{R}^K$ to classify InD data accurately in the logit space embedding $\mathbb{R}^K$. This DL model serves as a proxy for the feature space $\mathcal{X}$ to approximate the true feature map $\phi : \mathcal{X} \to \mathbb{R}^K$. Upon initialization, this DL model $f_\theta$ tends to project all data (both InD and OoD) toward the center of the logit space Komini & Girdzijauskas (2024). However, after training in $\mathcal{D}_{\text{InD}}$, these models learn to map InD data in distinct, nearly orthogonal class-wise groups $\mathbb{Z}_{\text{InD}} = \{\mathbb{Z}_k\}_{k=1}^{K}$ in the logit embedding space Komini & Girdzijauskas (2024); Dang et al. (2024). To do so, $f_\theta$ is optimized to identify and parameterize the optimal (as the model capacity allows) distribution of class-specific discriminative features of the InD data:

$$\theta^* \triangleq \arg \min_\theta \mathbb{E}_{(x,y) \sim \mathcal{D}_{\text{InD}}} \left[ - \log p_\theta(y|x) \right]. \tag{1}$$

It does this by maximizing the dot product of the model's parameters $\theta$ with optimally selected class-specific discriminative features, enhancing its ability to recognize similar patterns within each class Bishop (2006).

Thus, InD data ($x \in \mathcal{D}_{\text{InD}}$), characterized by class-specific features, interact coherently with the model's parameters through successive convolution layers. This compounding interaction leads to high positive values in the corresponding logit cell for the relevant class, while producing very low values in the other cells. Eventually, providing a linear separation between each class in the logit space embeddings.

Conversely, OoD data, which lack these class-specific features, result in all logit cells displaying very low output values Komini & Girdzijauskas (2024). This observation is crucial as it highlights the capability of DL models to create a separation between InD and OoD embeddings.

## B   Related Work

To the best of the authors' knowledge, this study introduces the first binary classification model that directly parameterizes a boundary distinguishing OoD from InD. Although theoretical investigations into OoD behavior exist Ye et al. (2021); Fang et al. (2022), it is important to highlight that current methodologies predominantly concentrate on creating effective scoring functions. These functions are tailored to assign high likelihoods to InD data, enabling dependable differentiation from OoD data.

a) A significant body of research leverages outputs from DL classifiers, such as logits or softmax probabilities, to construct scoring systems. These scoring mechanisms aim to measure the degree of model uncertainty, which serves as an indicator for identifying InD data. The most straightforward approaches might employ the maximum logit Hendrycks et al. (2022) or softmax output Hendrycks et al. (2019); Liang et al. (2018); Hsu et al. (2020) as the score. More sophisticated methods attempt to model the distribution of logits using parametric approaches, such as Gaussian distributions Lee et al. (2018); Ren et al. (2021), or energy-based models Liu et al. (2020). Additionally, some approaches employ ensemble techniques that improve the accuracy of uncertainty estimation by manipulating softmax output from several independent models, providing a more robust framework for distinguishing InD from OoD instances Vadera et al. (2020a); Lakshminarayanan et al. (2016); Malinin et al. (2020); Vadera et al. (2020b); Depeweg et al. (2018); Vadera et al. (2020a).

b) Alternatively, other studies exploit the gradients of the DL classifiers with respect to the input data, particularly when differentiating the model's output against a uniform probability array. In scenarios involving well-trained DL classifiers, InD data is expected to induce larger gradient magnitudes

than OoD data, a predictable outcome given that InD samples are optimized to yield non-uniform probabilities that maximize the logit cell, indicating the correct class Wu et al. (2024). Conversely, OoD data, not part of the training set, often leads the model to default to a uniform output for such samples. Building on this insight, the authors propose a novel scoring function based on the magnitude of the collective gradient across each model parameter to improve detection sensitivity for InD data.

c) Other approaches attempt to leverage intermediary features to create scoring functions. These features exhibit significant sensitivity to OoD data, with various methods employing techniques such as clipping Sun et al. (2021); Zhu et al. (2022); Djurisic et al. (2023); Xu et al. (2024); Yuan et al. (2024), measuring distances Park et al. (2023); Yu et al. (2023); Sun et al. (2022), or calculating norms Yu et al. (2023) to detect instances of OoD.

Despite the good performance reported for feature-based and gradient-based methods, these techniques generally demand more extensive tuning and engineering than logit-based methods. This is primarily because intermediary features carry less refined and more redundant InD-specific discriminatory features than logits, which provide optimal class distinction.

It should be noted that the techniques discussed herein are primarily models designed for InD data identification that have been adapted for OoD data detection by implementing a threshold-based model. These methods, however, do not involve a parameterized delineation of the boundary between InD and OoD data. In contrast, our research aims to explicitly parameterize a boundary that distinguishes between InD and OoD logits, enabling validation and testing.

## C SCORING-BASED METHODS FOR OoD DETECTION

Most existing scoring-based methods for OoD detection operate under the principle that *InD samples should receive higher scores than OoD samples*.

Let $f_\theta : \mathbb{R}^d \to \mathbb{R}^K$ denote a neural network classifier trained on $K$ classes, where an input $x \in \mathbb{R}^d$ is mapped to logits $f_\theta(x) \in \mathbb{R}^K$. We denote by $h(x) \in \mathbb{R}^N$ the embedding from the penultimate (pre-logit) layer, with $N \geq K$.

**Maximum Softmax Probability (MSP).** The MSP baseline Hendrycks & Gimpel (2017) assumes that the softmax distribution reflects the model's predictive uncertainty. The maximum predicted class probability is used as the InD score:

$$\text{MSP}(x) = \max_{i \in \{1, \ldots, K\}} \text{softmax}(f_\theta(x))_i.$$

**ODIN.** ODIN Liang et al. (2018) improves upon MSP by applying *temperature scaling* to the logits and introducing *input perturbations* designed to amplify the separation between InD and OoD. For a perturbed input $x_a$ and temperature $T > 0$, the score is:

$$\text{ODIN}(x) = \max_i \text{softmax}\left(\frac{f_\theta(x_a)}{T}\right)_i.$$

**GODIN.** GODIN Hsu et al. (2020) extends ODIN by explicitly decomposing the model output into class probability and InD probability, and optimizes perturbations to enhance OoD detection.

**Mahalanobis Distance.** Lee et al. Lee et al. (2018) proposed leveraging intermediate features at multiple layers. For each layer $l$, the class-conditional Gaussian distribution is estimated with mean $\mu_i^l$ and covariance $\Sigma^l$. The Mahalanobis-based score at layer $l$ is:

$$\text{Maha}(x)_l = \max_i -\left(f^l(x) - \mu_i^l\right)^\top \left(\Sigma^l\right)^{-1} \left(f^l(x) - \mu_i^l\right),$$

and the overall score is a weighted combination:

$$\text{Maha}(x) = \sum_l \alpha_l \, \text{Maha}(x)_l,$$

where the coefficients $\alpha_l$ are learned via logistic regression using InD versus adversarial examples.

**Energy-Based Score.**   Liu et al. Liu et al. (2020) argued that the softmax probability can be poorly calibrated for OoD detection. Instead, they proposed the *energy score*, defined as:

$$\text{Energy}(x) = -\log \sum_{i=1}^{K} \exp\big(f_\theta(x)_i\big),$$

where lower energy corresponds to a higher likelihood of being InD.

**K-Nearest Neighbors (KNN).**   KNN-based methods Sun et al. (2022) operate directly in the feature space. The OoD score is derived from the (negative) cumulative distance to the $K$ nearest neighbors of the InD embedding $h(x)$ in the $L_2$-normalized feature space:

$$\text{KNN}(x) = -\sum_{j=1}^{K} \big\| h(x) - h(x_j^{\text{NN}}) \big\|_2.$$

## D   GENERATING InD AND OoD EMBEDDINGS

---
**Algorithm 1** Generating InD and OoD Logit Embeddings
---

**Require:** InD dataset $\mathcal{D}_{\text{InD}} = \{(\mathbf{x}_i, y_i)\}_{i=1}^{N} = \cup_{k=1}^{K} \mathcal{C}_k$ with $K$ classes
**Require:** Auxiliary classifier $f_\theta^{\text{Aux}}$, main classifier $f_\theta$
**Ensure:** Combined InD and OoD logits $\mathbb{Z}$ along with its labels $\mathfrak{O}$

1: **procedure** LOGITGENERATION($\mathcal{D}_{\text{InD}}, f_\theta^{\text{Aux}}, f_\theta$)
2:      $\mathcal{D}_{\text{train}}^{\text{aux}} \leftarrow \bigcup_{k=1}^{K-1} \mathcal{C}_k$                                      ▷ Use first $K-1$ classes for auxiliary training
3:      Train $f_\theta^{\text{Aux}}$ on $\mathcal{D}_{\text{train}}^{\text{aux}}$                                      ▷ Learn on subset of classes
4:      $\mathcal{D}_{\text{OoD}}^{\text{aux}} \leftarrow \mathcal{D}_{\text{InD}} \setminus \mathcal{D}_{\text{train}}^{\text{aux}}$                                      ▷ Remaining class serves as OoD
5:      $\mathbb{Z}_{\text{OoD}}^{\text{aux}} \leftarrow f_\theta^{\text{Aux}}(\mathcal{D}_{\text{OoD}}^{\text{aux}})$                                      ▷ $(N_{\text{OoD}} \times (K-1))$ logits
6:      $\mathbb{Z}_{\text{OoD}}^{\text{impute}} \sim \text{Uniform}(\min \mathbb{Z}_{\text{OoD}}^{\text{aux}}, \max \mathbb{Z}_{\text{OoD}}^{\text{aux}})$                                      ▷ Sample missing dimension
7:      $\mathbb{Z}_{\text{OoD}} \leftarrow [\mathbb{Z}_{\text{OoD}}^{\text{aux}} \| \mathbb{Z}_{\text{OoD}}^{\text{impute}}]$                                      ▷ Concatenate to $(N_{\text{OoD}} \times K)$
8:      Train $f_\theta$ on full $\mathcal{D}_{\text{InD}}$                                      ▷ Learn on all $K$ classes
9:      $\mathbb{Z}_{\text{InD}} \leftarrow f_\theta(\mathcal{D}_{\text{InD}})$                                      ▷ $(N \times K)$ in-distribution logits
10:     $\mathbb{Z} \leftarrow [\mathbb{Z}_{\text{InD}} \| \mathbb{Z}_{\text{OoD}}]$                                      ▷ Combine all logits
11:     $\mathfrak{O} \leftarrow [\underbrace{0, \ldots, 0}_{N_{\text{InD}}} \| \underbrace{1, \ldots, 1}_{N_{\text{OoD}}}]$                                      ▷ Binary OoD labels
12:     **return** $(\mathbb{Z}, \mathfrak{O})$                                      ▷ Return logits and labels
13: **end procedure**

---

# E   TRAINING OF CLASSIFIER ON IND AND OOD EMBEDDINGS

Traditional DL models (i.e. $f_\theta : \mathcal{X} \to \mathbb{R}^K$) that address InD data classification typically enhance the likelihood values for logit regions associated with the correct class ($k$) and decrease those for others, achieving:

$$f_\theta(x)_k \gg f_\theta(x)_j \quad \forall j \neq k \tag{2}$$

This *relative* likelihood adjustment enables threshold-free decision-making, as predictions rely solely on the $\arg\max$ operation:

$$\hat{y} \triangleq \arg\max_{k \in \{1, \ldots, K\}} f_\theta(x)_k \tag{3}$$

We exploit the spatial displacement between the OoD and InD logits to adopt this paradigm for OoD detection. Notably, the class-wise clusters of InD logits exhibit near orthogonality and maintain a consistent displacement from the logit space center, in contrast to OoD logits, which tend to cluster towards this center Komini & Girdzijauskas (2024); Dang et al. (2024). Based on these spatial characteristics, we train the likelihood estimation model to assign high InD likelihood values to regions with densely concentrated InD logits and high OoD likelihood values to regions predominantly containing OoD logits (see fig. 1b).

Given that both InD and OoD logits (i.e., $\mathbb{Z} = [\mathbb{Z}_{\text{InD}} \cup \mathbb{Z}_{\text{OoD}}^{aux}]$, $\mathfrak{O} = [0, \ldots, 0, 1, \ldots, 1]$) reside in a known vector space that we can parameterize via its spanning orthogonal axis, it allows us directly parameterize the separation boundary for OoD and InD ($g_\eta : \mathbb{R}^K \to \mathbb{R}^2$ see fig. 1b) such as:

$$\eta^* \triangleq \arg\min_\eta \mathbb{E}_{\mathfrak{o} \sim \mathfrak{O}, z \sim \mathbb{Z}} [-\log p_\eta(\mathfrak{o}|z)] \quad \text{s.t: } \mathfrak{o} \text{ indicates whether is InD } (\mathfrak{o} = 0) \text{ or OoD } (\mathfrak{o} = 1).$$

Moreover, because the auxiliary OoD embeddings $\mathbb{Z}_{\text{OoD}}^{\text{aux}}$ and the InD embeddings $\mathbb{Z}_{\text{InD}}$ are expected to be well separated, we can directly validate both the quality of the generated auxiliary OoD embeddings and the training of the binary classifier $g_\eta : \mathbb{R}^K \to \mathbb{R}^2$.

This *relative* likelihood adjustment enables threshold-free decision-making, as predictions rely solely on the $\arg\max$ operation just as in the classification of $\mathcal{D}_{\text{InD}}$ as:

$$\hat{\mathfrak{o}} \triangleq \arg\max_{j \in \{0,1\}} g_\eta(z)_j \tag{4}$$

---

**Algorithm 2** Training of Binary Classifier on InD and OoD Logit Embeddings

---

**Require:** Combined logits $\mathbb{Z}$, labels $\mathfrak{O}$, initial parameters $\eta$, learning rate $\alpha$, iterations $N$
**Ensure:** Trained binary classifier $g_\eta : \mathbb{R}^K \to \mathbb{R}^2$
 1: **procedure** TRAINBINARYCLASSIFIER($\mathbb{Z}, \mathfrak{O}, g_\eta$)
 2:   **for** $i = 1, \ldots, N$ **do**                                  ▷ Iterate for $N$ epochs
 3:     **for** $(\mathfrak{o}, z) \in (\mathfrak{O}, \mathbb{Z})$ **do**                    ▷ For each (label, logit) pair
 4:       $[\ell^{\text{InD}}, \ell^{\text{OoD}}] \leftarrow g_\eta(z)$                      ▷ Compute predictions
 5:       $\sigma^{\text{InD}}, \sigma^{\text{OoD}} \leftarrow \text{softmax}(\ell^{\text{InD}}, \ell^{\text{OoD}})$
 6:       $\mathcal{L}(\eta) \leftarrow -(1 - \mathfrak{o})\log\sigma^{\text{InD}} - \mathfrak{o}\log\sigma^{\text{OoD}}$        ▷ Cross-entropy loss
 7:       $\eta \leftarrow \eta - \alpha\nabla_\eta\mathcal{L}(\eta)$                   ▷ Update parameters via gradient descent
 8:     **end for**
 9:   **end for**
 10:   **return** $g_\eta$
 11: **end procedure**

---

# F  ANALYTICAL ANALYSIS

By Lemma 1, excluding a single class $k$ and using its excluded-class proxy as auxiliary OoD embeddings yields an approximation error of order $\mathcal{O}(\sigma(C_k))$ (i.e., the upperbound of the approximation error of $f_\theta^{\text{Aux}}$ relative to $f_\theta$). If instead we exclude a set $S$ of $N > 1$ classes, a straightforward perturbation argument gives a bound of order

$$\mathcal{O}\left(\sum_{j \in S} \sigma(C_j)\right),$$

which is necessarily larger—and therefore looser—than the best single–class bound. In particular, given $\sigma(C_j) \geq 0, \forall j \in \{1, \ldots, K\}$, we have

$$\mathcal{O}\big(\sigma(C_k)\big) \ \leq \ \mathcal{O}\left(\sum_{j \in S} \sigma(C_j)\right) \quad \text{for any } S \subseteq \{1, \ldots, k, \ldots, K\},\ |S| = N > 1.$$

Hence, the tightest provable approximation is obtained with $N = 1$.

**Assumption 1.** *Given the training objective (see eq. (1)) is to parameterize the optimal representation of class-specific features from the training data (i.e., $\mathcal{D}_{InD} = \cup_{k=1}^K \mathcal{C}_k$), their per class discrimination is possible if their class-specific features are mutually independent (i.e., $\mathcal{C}_i \perp\!\!\!\perp \mathcal{C}_j, \forall i \neq j$).*

**Lemma 1** (Auxiliary-model bound via score variance). *Let $f_\theta : \mathcal{X} \rightarrow \mathbb{R}^K$ be the main classifier and $f_\theta^{\text{Aux}}$ an auxiliary approximation to $f_\theta$. For every single class $k$, define the class-wise score-variance*

$$\sigma(C_k) \ \triangleq \ \text{Var}\big(\nabla_\theta \log p_\theta(y = k \mid x)\big).$$

*Assume that for samples $(x, y = k)$ and small parameter perturbations $\delta\theta$,*

$$\big\|f_{\theta+\delta\theta}(x) - f_\theta(x)\big\|_2 \ \leq \ C\,\sigma(C_k),$$

*for some constant $C > 0$. Then the point-wise approximation error satisfies*

$$\big\|f_\theta^{\text{Aux}}(x) - f_\theta(x)\big\|_2 \ \leq \ \mathcal{O}(\sigma(C_k)).$$

*Proof.* To analyze feature perturbations $\Delta f$, we examine the Kullback-Leibler divergence between the model distributions before and after parameter perturbation $\delta\theta$:

$$\text{KL}(p_\theta \parallel p_{\theta+\delta\theta}) = \mathbb{E}_{x \sim \mathcal{D}_{\text{InD}}/\mathcal{C}_k}\left[\log \frac{p_\theta(y|x)}{p_{\theta+\delta\theta}(y|x)}\right]$$

A Taylor expansion around $\delta\theta = 0$ :

- The gradient of KL divergence at $\delta\theta = 0$ equals zero:

$$\nabla_{\delta\theta} \text{KL}(p_\theta \parallel p_{\theta+\delta\theta})\big|_{\delta\theta=0} = -\mathbb{E}\left[\nabla_\theta \log p_\theta(y|x)\right] = 0$$

- The Hessian gives the FIM:

$$\begin{aligned}
\nabla_{\delta\theta}^2 \text{KL}(p_\theta \parallel p_{\theta+\delta\theta})\big|_{\delta\theta=0} &= -\mathbb{E}_{x \sim \mathcal{D}_{\text{InD}}/\mathcal{C}_k}\left[\nabla_\theta^2 \log p_\theta(y|x)\right] \\
&= \mathbb{E}_{x \sim \mathcal{D}_{\text{InD}}/\mathcal{C}_k}\left[(\nabla_\theta \log p_\theta(y|x))(\nabla_\theta \log p_\theta(y|x))^\top\right] \\
&= \mathcal{I}_{\mathcal{D}_{\text{InD}}/\mathcal{C}_k}(\theta)
\end{aligned}$$

This yields the second-order approximation:

$$\text{KL}(p_\theta \parallel p_{\theta+\delta\theta}) \approx \frac{1}{2}\delta\theta^\top \mathcal{I}_{\mathcal{D}_{\text{InD}}/\mathcal{C}_k}(\theta)\delta\theta + \mathcal{O}(\|\delta\theta\|^3)$$

This approximation reveals that:

- the FIM $I(\theta)$ acts as a Riemannian metric on the parameter space, defining the "distance" between $p_\theta$ and $p_{\theta+\delta\theta}$.

$$\mathrm{KL}(p_\theta \parallel p_{\theta+\delta\theta}) \propto \delta\theta^\top \mathcal{I}_{\mathcal{D}_{\mathrm{InD}}/\mathcal{C}_k}(\theta)\delta\theta.$$

- for small $\delta\theta$, the change in features $f_\theta(x)$ is thus:

$$\|f_{\theta+\delta\theta}(x) - f_\theta(x)\| \propto \delta\theta^\top \mathcal{I}_{\mathcal{D}_{\mathrm{InD}}/\mathcal{C}_k}(\theta)\delta\theta.$$

The exclusion of a single class induces a smaller perturbation to the FIM (i.e., $\|\mathcal{I}_{\mathcal{D}_{\mathrm{InD}}}(\theta) - \mathcal{I}_{\mathcal{D}_{\mathrm{InD}}/\mathcal{C}_k}(\theta)\|_2 = \sigma(C_k)$ see Lemma 2), compared to excluding multiple classes. This reduced perturbation leads to better preservation of the Riemannian metric structure (i.e., $\delta\theta^\top I(\theta)\delta\theta$). Consequently, the resulting feature displacement under parameter updates $\delta\theta$ remains tightly bounded:

$$\|f_{\theta+\delta\theta}(x) - f_\theta(x)\|_2 \leq \mathcal{O}(\sigma(C_k))$$

In the case of multiple class exclusions that bound increases to $\sum_{i=k}^{K} \mathcal{O}(\sigma(C_i))$ $\qquad\square$

**Lemma 2.** *[Fisher Stability of Single-Class Exclusion] Let $L(\theta)$ be the cross-entropy loss, and $I(\theta)$ the Fisher Information Matrix (FIM) of the full dataset $\mathcal{D}_{InD} = \cup_{k=1}^{K}\mathcal{C}_k$. When excluding a single class $C_k$, the perturbation to the FIM is bounded by:*

$$\|\mathcal{I}_{\mathcal{D}_{InD}}(\theta) - \mathcal{I}_{\mathcal{D}_{InD}/\mathcal{C}_k}(\theta)\|_2 = \sigma(C_k)$$

*where $\sigma(C_k)$ is the variance of the score functions (gradients of log-likelihood) for data $x \sim \mathcal{C}_k$.*

*Proof.* For a parametric model $p_\theta(y \mid x)$ with parameters $\theta$, the Fisher Information Matrix (FIM)

$$\mathcal{I}(\theta) = \mathbb{E}_{x,y \sim \mathcal{D}_{\mathrm{InD}}} \left[ \nabla_\theta \log p_\theta(y \mid x)\nabla_\theta \log p_\theta(y \mid x)^\top \right],$$

quantifies how sensitive the likelihood is to parameter changes by estimating the covariance matrix of the score function (gradient of log-likelihood):

Notice that for two independent datasets $X$ and $Y$ the FIM is additive:

$$\mathcal{I}_{X \cup Y}(\theta) = \mathcal{I}_X(\theta) + \mathcal{I}_Y(\theta). \tag{5}$$

Under the assumption of class-conditional independence (i.e., $\mathcal{C}_i \perp\!\!\!\perp \mathcal{C}_j, \forall i \neq j$), we can use the additive property of FIM (see eq. (5)), and then we can decompose the FIM as follows:

$$\mathcal{I}_{\mathcal{D}_{\mathrm{InD}}}(\theta) = \mathcal{I}_{\mathcal{D}_{\mathrm{InD}}/\mathcal{C}_k}(\theta) + \mathcal{I}_{\mathcal{C}_k}(\theta).$$

Under cross-entropy loss, this simplifies to the gradient covariance matrix.

When excluding class $C_k$, the FIM becomes:

$$\begin{aligned} \mathcal{I}_{\mathcal{C}_k}(\theta) &= \mathbb{E}_{x \sim \mathcal{C}_k} \left[ \nabla_\theta \log p_\theta(y = k \mid x)\nabla_\theta \log p_\theta(y = k \mid x)^\top \right] \\ &= \mathbb{E}_{x \sim \mathcal{C}_k} \left[ \|\nabla_\theta \log p_\theta(y = k \mid x)\|_2^2 \right] \\ &\triangleq \mathrm{Var}\left( \nabla_\theta \log p_\theta(y = k \mid x) \right) \\ &= \sigma(C_k), \end{aligned}$$

where $\sigma(C_k)$ is the variance of the score functions (gradients of log-likelihood) of the data drawn from $x \sim \mathcal{C}_k$.

The step $\triangleq$ follows from $\mathbb{E}\left[ \nabla_\theta \log p_\theta(y|x) \right] = 0$.

The spectral norm of the perturbation is bounded by:

$$\|\Delta\mathcal{I}\|_2 = \|\mathcal{I}_{\mathcal{D}_{\mathrm{InD}}}(\theta) - \mathcal{I}_{\mathcal{D}_{\mathrm{InD}}/\mathcal{C}_k}(\theta)\|_2 = \|\mathcal{I}_{\mathcal{C}_k}(\theta)\|_2 - \sigma(C_k).$$

Notice that for multiple classes $\{k, \cdots, K\}$ exclusion the bound increases to

$$\|\Delta I\|_2 = \sum_{i=k}^{K} \sigma(C_i)$$

□

## G  ANALYSIS ACROSS DIFFERENT CLASS EXCLUSION

A natural question is: why not simply collect arbitrary data outside the training domain? While such data will produce OoD embeddings, there is no guarantee that they define a representative decision boundary relative to the InD distribution. In contrast, a *left-out* class from the same labeled dataset is known (by annotation) to be semantically distinct from the training classes, yet it often shares non-class-specific factors (e.g., sensor characteristics, low-level statistics, augmentation pipeline). This combination—different class-specific features with similar nuisance factors—yields *near-OoD* examples that more faithfully probe the InD boundary and thus provide a more representative separation criterion than arbitrary out-of-domain samples. Given that excluding a single class is the most effective way to approximate OOD data, one may wonder how to properly select that class. A perturbation analysis over all choices of the held-out class indicates that the approximation quality is essentially invariant to which class is excluded; the resulting auxiliary-OoD logits show negligible variation between choices (see fig. 5). Moreover, the classification accuracy on the stays stable across different scenarios (see fig. 4).

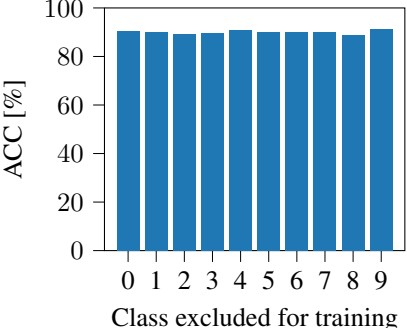

Figure 4: OoD detection performance as the classifier is trained with an increasing number of classes—starting from a binary model, then a ternary model, up to all ten classes. In each setting, evaluation uses the InD embeddings from the corresponding classifier from CIFAR-10, while SVHN is used as OODs.

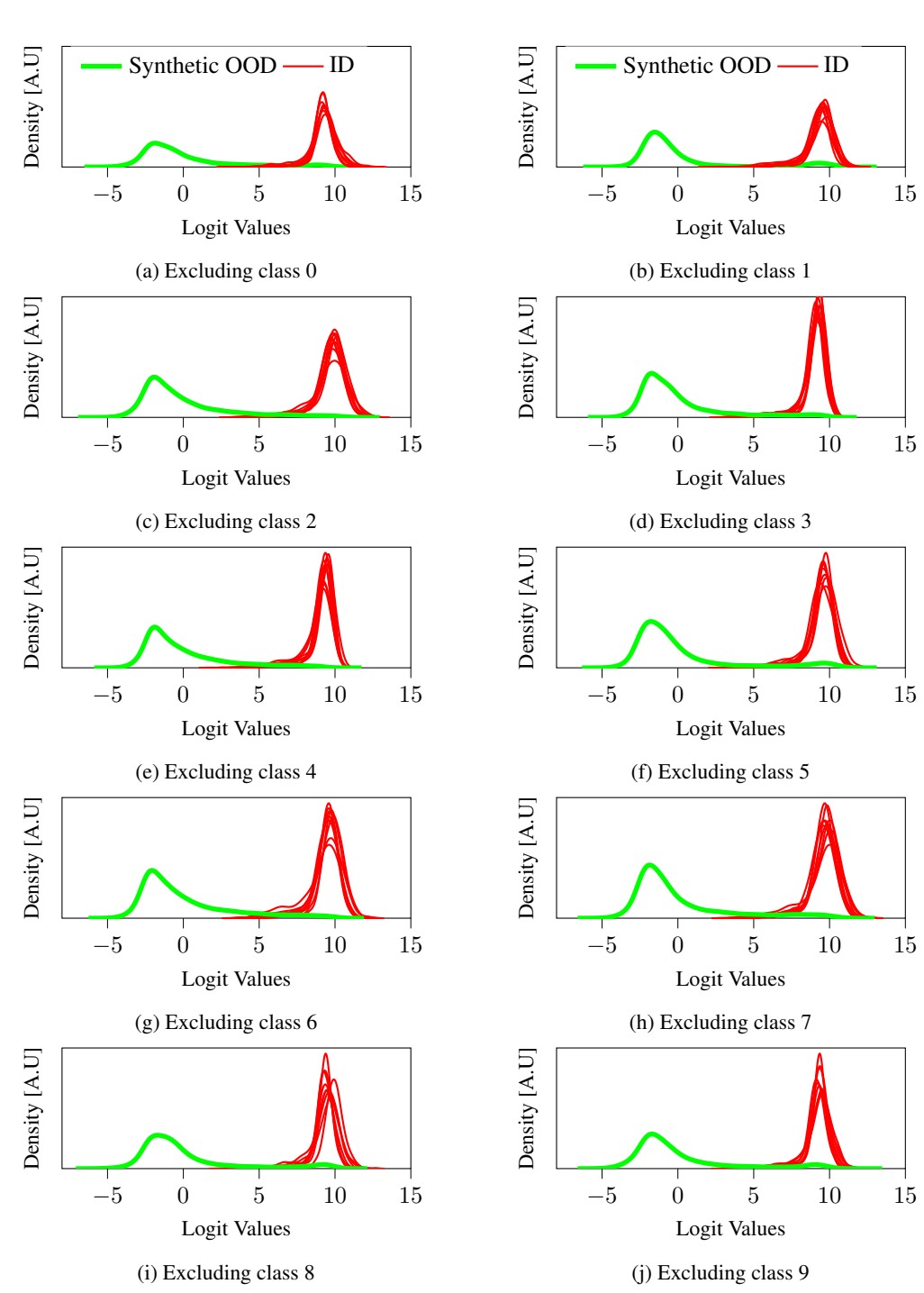

Figure 5: Auxiliary OoD logit embeddings when excluding individual classes from CIFAR-10 during training with ResNet-18. Each subfigure corresponds to a separate experiment in which one of the ten classes is held out (from class 0 in fig. 5a to class 9 in fig. 5j). Across all cases, the auxiliary OoD distributions remain highly consistent, supporting the class-invariance of our approach.

# H  ANALYSIS ACROSS DIFFERENT NUMBERS OF CLASS INCLUDED IN THE TRAINING

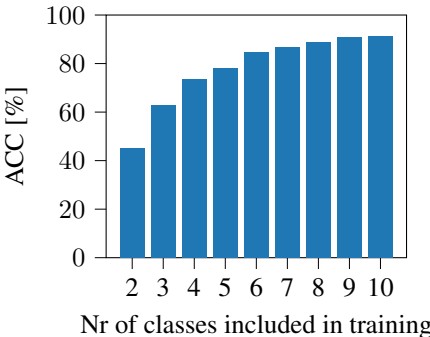

Figure 6: OoD detection performance under a leave-one-class-out protocol on CIFAR-10, where each experiment trains the classifier on 9 classes and excludes a different single class.

As the number of classes used to train the classifier increases, not only do the OoD approximations improve, but the classifier's ability to separate OoD from InD embeddings also strengthens. Empirically, fig. 7 shows that adding more InD classes yields logits with larger and more uniform displacement from the logit-space center, producing clearer separation (disentanglement) between InD and OoD logits. Conversely, when the classifier is trained on only a subset of classes, OoD detection performance degrades (see fig. 6). Even when InD classification accuracy is held fixed across study cases, OoD separability improves as more training data (classes) are included. This indicates that as the number of classes increases, the learned logit representations become more class-specific, sharpening the InD decision structure and reducing the model's propensity to form spurious correlations with OoD data.

Notice that our approach assumes the classifier operates on a sufficiently large and balanced set of InD classes. When the number of classes is small, the classifier's ability to distinguish between InD and OoD embeddings deteriorates, and the generation of reliable auxiliary OoD samples becomes unstable. Furthermore, our method does not explicitly account for class imbalance. If certain classes are underrepresented, their embeddings tend to lie closer to the feature space center, resulting in an uneven distribution of displacements across classes. This imbalance not only weakens the approximation of OoD embeddings but also hinders the classifier's capacity to effectively disentangle OoD from InD samples along those dimensions.

To assess the tightness of our approximation, we conduct eight controlled experiments on CIFAR-10 using a ResNet-34, while SVHN is used as the OoD. In each run, we train on a subset of the label space with 2 to 9 classes (i.e., excluding 8 or 1 classes, respectively). We observe a sharp transition: training with nine classes—excluding only one—yields behavior that closely matches that of the model trained on all ten classes, whereas removing two or more classes produces markedly larger deviations.

Additionally, the DL classifier is central to separating OoD from InD embeddings. The more effectively the DL model learns to parameterize the class-specific discriminative features in the InD dataset, the better it can partition InD from OoD in embedding space. Under a standard classification objective over $K$ classes, each labeled example contributes gradients that (i) reinforce the correct class logit and (ii) suppress the $K - 1$ competing logits, thereby enlarging inter-class margins in the embedding space Wu et al. (2018); Sun et al. (2017); Deng et al. (2019). These gradients solely drive the updates to the parameters that encode the optimal class-specific features present in the training set. Consequently, we get increase the amount of gradients in two complementary ways: (1) *more labeled examples per class* reduce variance in the gradient estimates and yield more representative class features; and (2) *a larger number of classes* increases the number of negative contrasts per update, suppressing non-class specific features that exist among a broader set of alternatives. We distinguish these effects: more data per class improves how precisely each class is learned, whereas increasing the number of classes refines the class-specific discriminative features by forcing the model to disentangle finer distinctions among classes (see fig. 6 for more details).

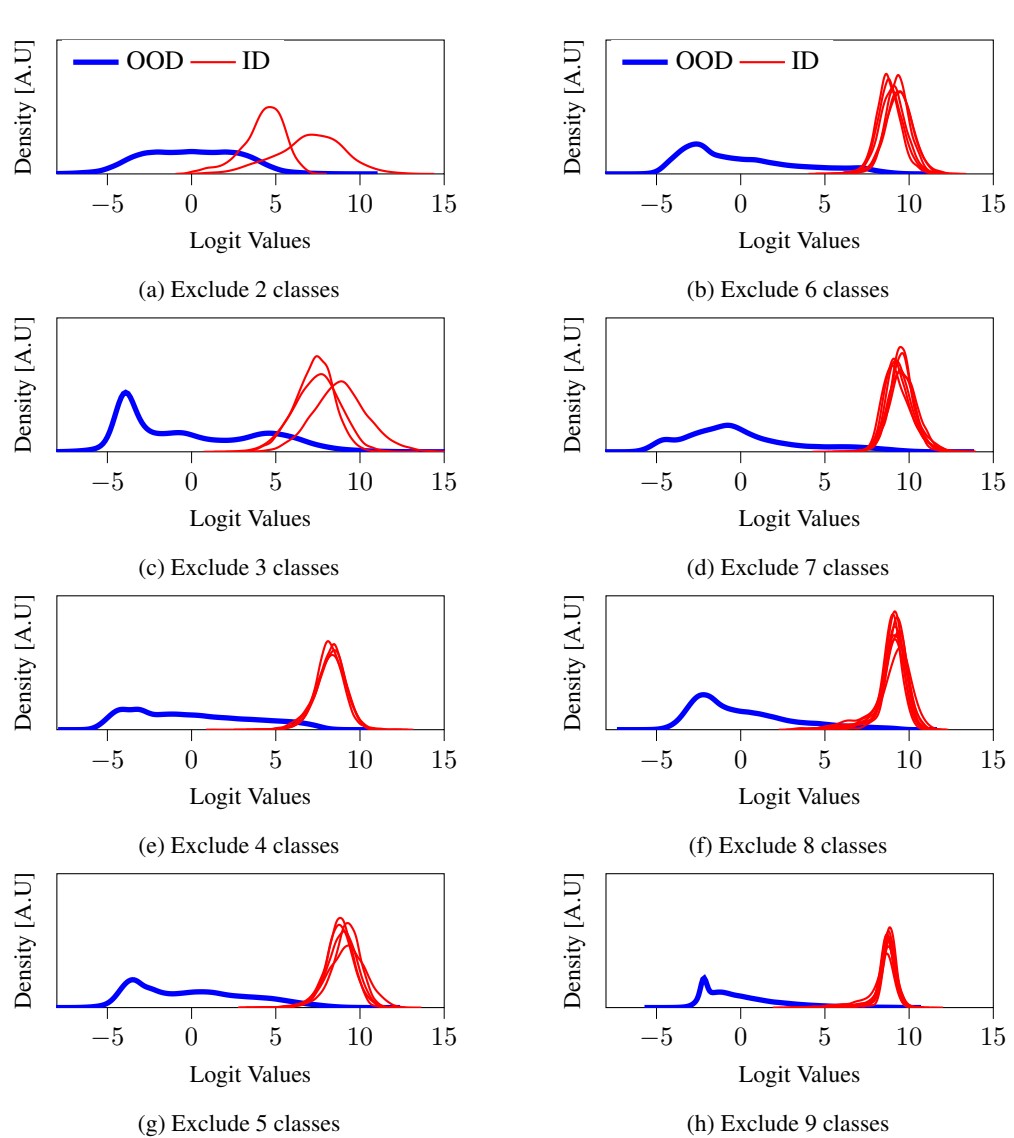

Figure 7: **Logit behavior under class-exclusion ablations (ResNet-34, CIFAR-10).** Each panel shows the logits when training excludes $k$ classes, with $k \in \{2, \ldots, 9\}$ (panels a–h correspond to $k = 2$ through $k = 9$).

# I    Synthetic OoD across different architectures

**Ablations.**    We conduct two ablations to assess robustness: (a) across different architectures and (b) across the choice of in-distribution (InD) dataset. For each model (ResNet and DenseNet variants), we train on an InD dataset and fit a lightweight binary classifier $g_\eta : \mathbb{R}^K \to \mathbb{R}^2$ on top of the logits, using InD and auxiliary-OoD logits. This yields both an InD score $\ell^{\mathrm{InD}}(z \mid \eta)$ and an OoD score $\ell^{\mathrm{OoD}}(z \mid \eta)$.

**Metrics.** Following common practice, we report AUROC computed from $\ell^{\mathrm{InD}}(z \mid \eta)$ (see table 1). To evaluate cross-architecture consistency, we additionally report InD/OoD classification accuracy (ACC) computed from $\ell^{\mathrm{OoD}}(z \mid \eta)$ (see table 2).

**Protocols.** (i) *Architectures:* we train multiple ResNet and DenseNet variants on CIFAR-10 and evaluate both $\ell^{\mathrm{InD}}$ and $\ell^{\mathrm{OoD}}$ via the auxiliary classifier, summarizing ACC across models in table 2. (ii) *InD dataset swap:* we repeat the procedure with SVHN as the InD dataset, reporting ACC in table 3. Density plots from the auxiliary model and the fully trained model are provided in fig. 3 and Appendix I.

For all three architectures, the same experimental protocol is followed: logit densities are visualized for in-distribution (CIFAR-10) and OoD(synthetic and SVHN) data, using both the auxiliary and full classifiers.

Table 2: InD and OoD accuracy across classification backbones trained with CIFAR-10 as the InD dataset.

| method | SVHN | | LSUN | | iSUN | | Textures | | Places365 | |
| --- | --- | --- | --- | --- | --- | --- | --- | --- | --- | --- |
| | ACC↑ | AUROC↑ | ACC↑ | AUROC↑ | ACC↑ | AUROC↑ | ACC↑ | AUROC↑ | ACC↑ | AUROC↑ |
| ResNet-18 | 91.31 | 98.79 | 91.71 | 98.90 | 92.01 | 98.99 | 81.01 | 95.13 | 82.10 | 96.32 |
| ResNet-34 | 91.19 | 98.11 | 91.11 | 98.23 | 91.71 | 98.14 | 82.12 | 94.21 | 81.73 | 95.27 |
| ResNet-50 | 91.43 | 98.39 | 91.76 | 98.11 | 91.31 | 98.79 | 82.04 | 94.32 | 81.12 | 95.18 |
| ResNet-101 | 90.63 | 98.13 | 91.54 | 98.01 | 91.43 | 98.14 | 82.01 | 95.56 | 81.01 | 95.67 |
| ResNet-152 | 90.69 | 98.68 | 91.98 | 98.73 | 92.98 | 98.99 | 82.12 | 95.42 | 80.13 | 95.15 |
| DenseNet-121 | 91.78 | 98.32 | 91.11 | 98.11 | 91.82 | 98.54 | 81.89 | 95.13 | 80.99 | 95.71 |
| DenseNet-161 | 91.33 | 98.21 | 90.58 | 97.90 | 92.17 | 98.13 | 81.23 | 95.87 | 81.08 | 96.03 |
| DenseNet-169 | 91.52 | 98.04 | 91.32 | 98.12 | 91.93 | 97.99 | 82.11 | 95.21 | 81.13 | 95.11 |
| DenseNet-201 | 91.01 | 98.13 | 90.45 | 98.01 | 91.20 | 98.01 | 82.03 | 94.09 | 80.55 | 95.09 |

Table 3: InD and OoD accuracy across classification backbones trained with SVHN as the InD dataset.

| method | CIFAR-10 | | LSUN | | iSUN | | Textures | | Places365 | |
| --- | --- | --- | --- | --- | --- | --- | --- | --- | --- | --- |
| | ACC↑ | AUROC↑ | ACC↑ | AUROC↑ | ACC↑ | AUROC↑ | ACC↑ | AUROC↑ | ACC↑ | AUROC↑ |
| ResNet-18 | 88.82 | 98.18 | 88.95 | 98.72 | 88.09 | 98.24 | 80.52 | 93.34 | 80.72 | 95.82 |
| ResNet-34 | 88.21 | 98.22 | 88.63 | 98.34 | 88.16 | 98.67 | 80.03 | 93.75 | 80.19 | 95.14 |
| ResNet-50 | 88.89 | 98.38 | 88.99 | 98.13 | 88.81 | 98.82 | 80.63 | 93.81 | 80.62 | 95.72 |
| ResNet-101 | 88.91 | 98.63 | 88.84 | 98.32 | 88.47 | 98.91 | 80.49 | 93.18 | 80.83 | 95.23 |
| ResNet-152 | 87.77 | 98.56 | 88.84 | 98.81 | 88.36 | 98.04 | 80.37 | 93.64 | 80.91 | 94.14 |
| DenseNet-121 | 88.34 | 98.14 | 88.64 | 98.23 | 88.77 | 98.34 | 80.83 | 93.48 | 80.22 | 95.01 |
| DenseNet-161 | 88.54 | 98.23 | 88.12 | 98.42 | 88.63 | 98.52 | 80.17 | 93.28 | 80.44 | 95.13 |
| DenseNet-169 | 88.82 | 98.64 | 88.63 | 98.51 | 88.71 | 98.19 | 80.19 | 93.71 | 80.82 | 94.14 |
| DenseNet-201 | 88.26 | 98.63 | 88.19 | 98.13 | 88.82 | 98.84 | 80.62 | 93.33 | 80.53 | 94.31 |

## I.1    Evaluating ResNet on CIFAR-10 (InD) with SVHN as OoD

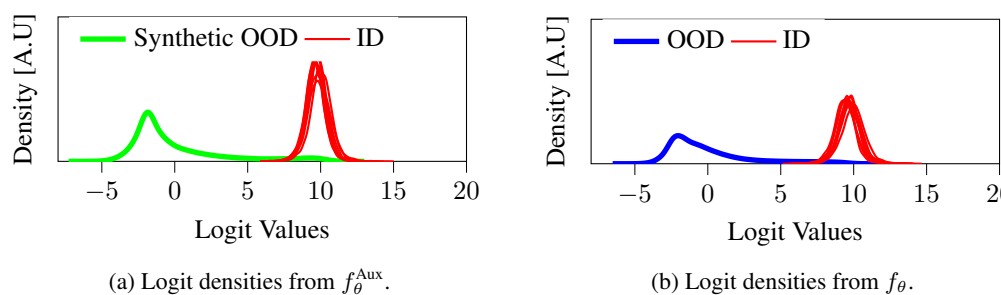

(a) Logit densities from $f_\theta^{\text{Aux}}$.

(b) Logit densities from $f_\theta$.

Figure 8: Logit density plots produced by the auxiliary classifier $f_\theta^{\text{Aux}}$ (trained on $K-1$ CIFAR-10 classes) and the full classifier $f_\theta$ (trained on all $K$ classes). Figure 8a presents InD logit distributions for CIFAR-10 classes 1–9 and synthetic OoD data (class 10), while fig. 8b displays distributions for all 10 classes as InD and SVHN as OoD. All results shown here use ResNet-18 as the backbone.

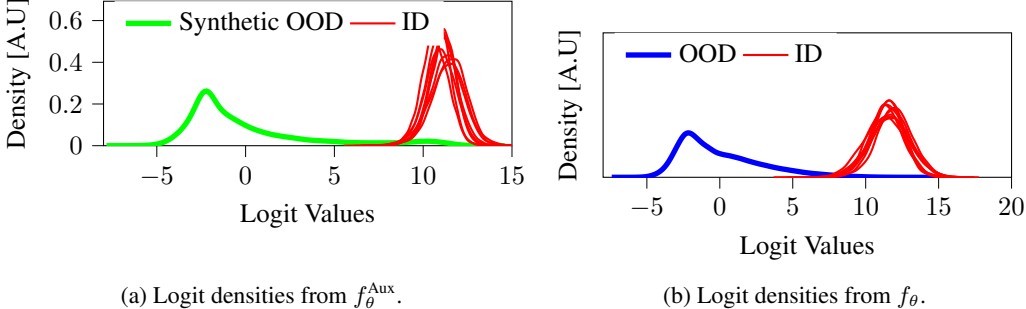

(a) Logit densities from $f_\theta^{\text{Aux}}$.

(b) Logit densities from $f_\theta$.

Figure 9: Logit density plots produced by the auxiliary classifier $f_\theta^{\text{Aux}}$ (trained on $K-1$ CIFAR-10 classes) and the full classifier $f_\theta$ (trained on all $K$ classes). Figure 9a presents InD logit distributions for CIFAR-10 classes 1–9 and synthetic OoD data (class 10), while fig. 9b displays distributions for all 10 classes as InD and SVHN as OoD. All results shown here use ResNet-50 as the backbone.

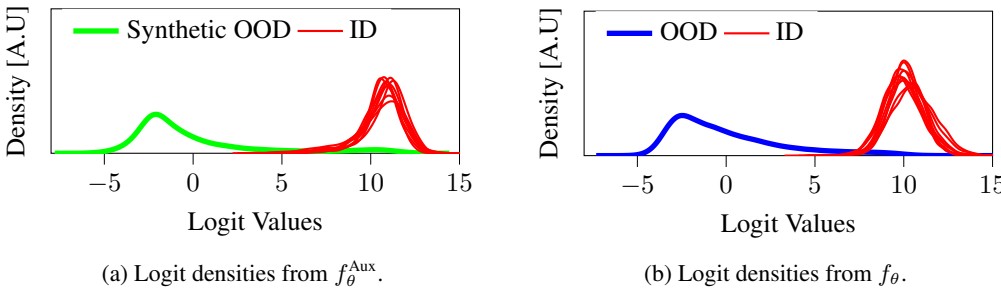

(a) Logit densities from $f_\theta^{\text{Aux}}$.

(b) Logit densities from $f_\theta$.

Figure 10: Logit density plots produced by the auxiliary classifier $f_\theta^{\text{Aux}}$ (trained on $K-1$ CIFAR-10 classes) and the full classifier $f_\theta$ (trained on all $K$ classes). Figure 10a presents InD logit distributions for CIFAR-10 classes 1–9 and synthetic OoD data (class 10), while fig. 10b displays distributions for all 10 classes as InD and SVHN as OoD. All results shown here use ResNet-101 as the backbone.

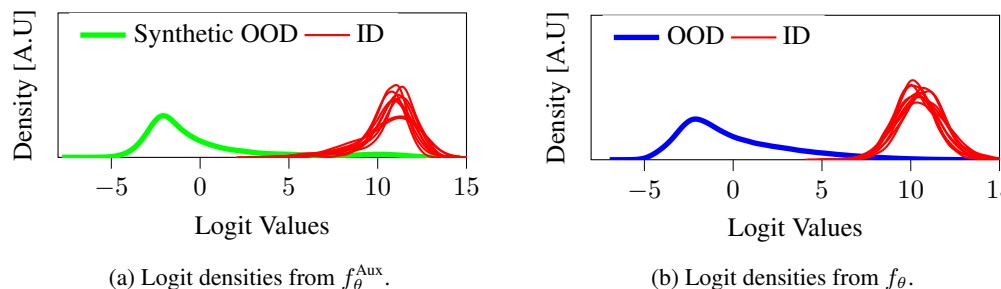

(a) Logit densities from $f_\theta^{\text{Aux}}$.

(b) Logit densities from $f_\theta$.

Figure 11: Logit density plots produced by the auxiliary classifier $f_\theta^{\text{Aux}}$ (trained on $K - 1$ CIFAR-10 classes) and the full classifier $f_\theta$ (trained on all $K$ classes). Figure 11a presents InD logit distributions for CIFAR-10 classes 1–9 and synthetic OoD data (class 10), while fig. 11b displays distributions for all 10 classes as InD and SVHN as OoD. All results shown here use ResNet-101 as the backbone.

## I.2 Evaluating ResNet on SVHN (InD) with CIFAR-10 as OoD

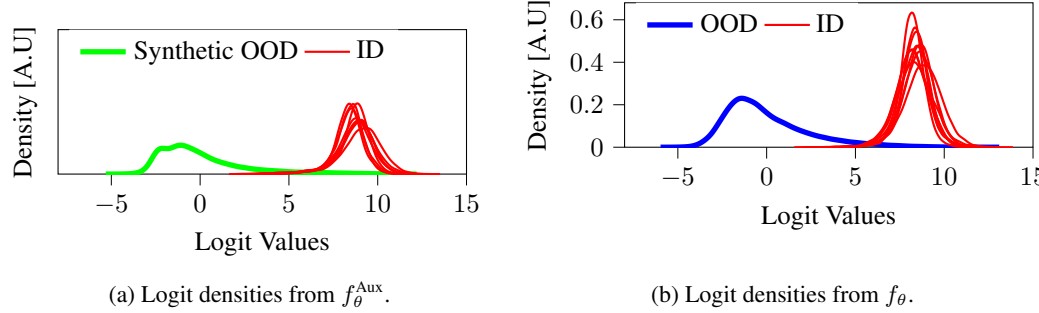

(a) Logit densities from $f_\theta^{\text{Aux}}$.

(b) Logit densities from $f_\theta$.

Figure 12: Logit density plots produced by the auxiliary classifier $f_\theta^{\text{Aux}}$ (trained on $K - 1$ CIFAR-10 classes) and the full classifier $f_\theta$ (trained on all $K$ classes). Figure 12a presents InD logit distributions for SVHN classes 1–9 and synthetic OoD data (class 10), while fig. 12b displays distributions for all 10 classes as InD and CIFAR-10 as OoD. All results shown here use ResNet-18 as the backbone.

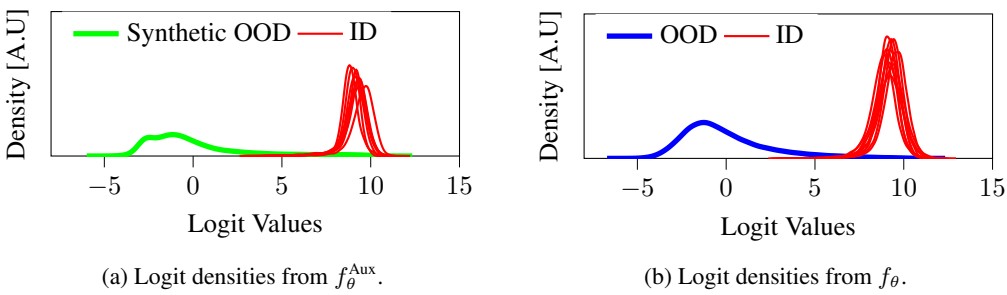

(a) Logit densities from $f_\theta^{\text{Aux}}$.

(b) Logit densities from $f_\theta$.

Figure 13: Logit density plots produced by the auxiliary classifier $f_\theta^{\text{Aux}}$ (trained on $K - 1$ CIFAR-10 classes) and the full classifier $f_\theta$ (trained on all $K$ classes). Figure 13a presents InD logit distributions for SVHN classes 1–9 and synthetic OoD data (class 10), while fig. 13b displays distributions for all 10 classes as InD and CIFAR-10 as OoD. All results shown here use ResNet-34 as the backbone.

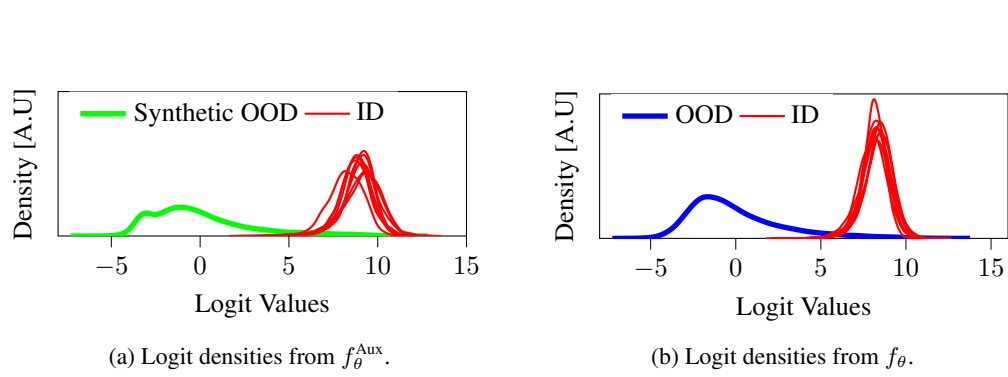

(a) Logit densities from $f_\theta^{\text{Aux}}$.

(b) Logit densities from $f_\theta$.

Figure 14: Logit density plots produced by the auxiliary classifier $f_\theta^{\text{Aux}}$ (trained on $K - 1$ CIFAR-10 classes) and the full classifier $f_\theta$ (trained on all $K$ classes). Figure 14a presents InD logit distributions for SVHN classes 1–9 and synthetic OoD data (class 10), while fig. 14b displays distributions for all 10 classes as InD and CIFAR-10 as OoD. All results shown here use ResNet-50 as the backbone.

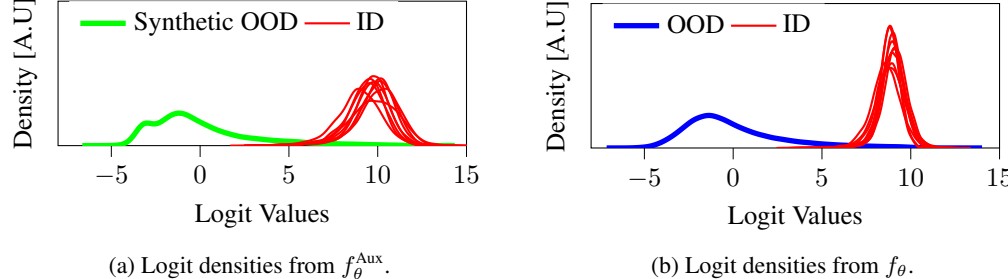

(a) Logit densities from $f_\theta^{\text{Aux}}$.

(b) Logit densities from $f_\theta$.

Figure 15: Logit density plots produced by the auxiliary classifier $f_\theta^{\text{Aux}}$ (trained on $K - 1$ CIFAR-10 classes) and the full classifier $f_\theta$ (trained on all $K$ classes). Figure 15a presents InD logit distributions for SVHN classes 1–9 and synthetic OoD data (class 10), while fig. 15b displays distributions for all 10 classes as InD and CIFAR-10 as OoD. All results shown here use ResNet-101 as the backbone.

I.3   EVALUATING DENSENET ON CIFAR-10 (IND) SVHN WITH AS OOD

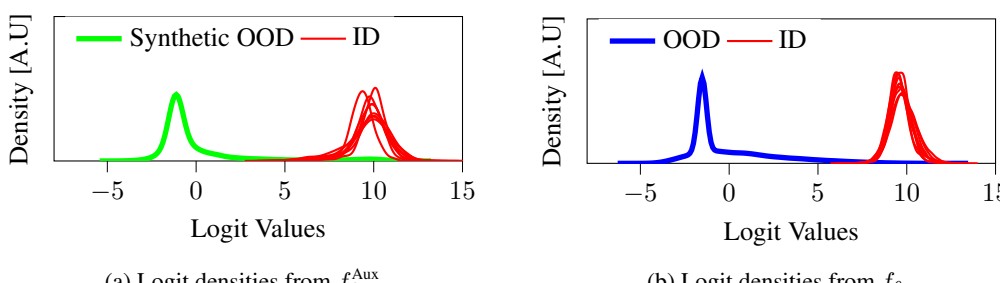

(a) Logit densities from $f_\theta^{\text{Aux}}$.       (b) Logit densities from $f_\theta$.

Figure 16: Logit density plots produced by the auxiliary classifier $f_\theta^{\text{Aux}}$ (trained on $K-1$ CIFAR-10 classes) and the full classifier $f_\theta$ (trained on all $K$ classes). Figure 16a presents InD logit distributions for CIFAR-10 classes 1–9 and synthetic OoD data (class 10), while fig. 16b displays distributions for all 10 classes as InD and SVHN as OoD. All results shown here use DenseNet-121 as the backbone.

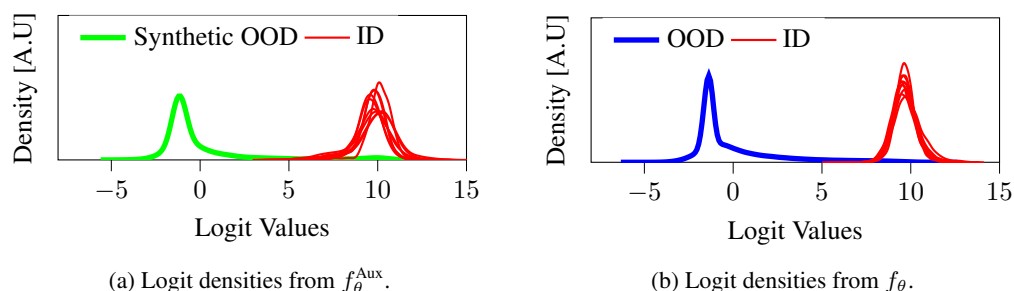

(a) Logit densities from $f_\theta^{\text{Aux}}$.       (b) Logit densities from $f_\theta$.

Figure 17: Logit density plots produced by the auxiliary classifier $f_\theta^{\text{Aux}}$ (trained on $K-1$ CIFAR-10 classes) and the full classifier $f_\theta$ (trained on all $K$ classes). Figure 17a presents InD logit distributions for CIFAR-10 classes 1–9 and synthetic OoD data (class 10), while fig. 17b displays distributions for all 10 classes as InD and SVHN as OoD. All results shown here use DenseNet-169 as the backbone.

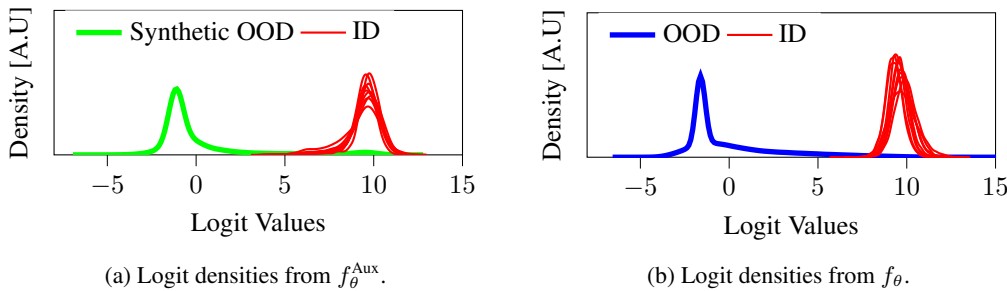

(a) Logit densities from $f_\theta^{\text{Aux}}$.       (b) Logit densities from $f_\theta$.

Figure 18: Logit density plots produced by the auxiliary classifier $f_\theta^{\text{Aux}}$ (trained on $K-1$ CIFAR-10 classes) and the full classifier $f_\theta$ (trained on all $K$ classes). Figure 18a presents InD logit distributions for CIFAR-10 classes 1–9 and synthetic OoD data (class 10), while fig. 18b displays distributions for all 10 classes as InD and SVHN as OoD. All results shown here use DenseNet-161 as the backbone.

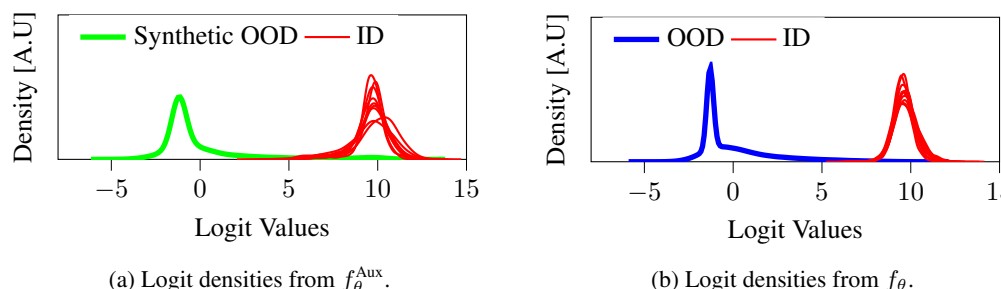

(a) Logit densities from $f_\theta^{\text{Aux}}$.

(b) Logit densities from $f_\theta$.

Figure 19: Logit density plots produced by the auxiliary classifier $f_\theta^{\text{Aux}}$ (trained on $K - 1$ CIFAR-10 classes) and the full classifier $f_\theta$ (trained on all $K$ classes). Figure 19a presents InD logit distributions for CIFAR-10 classes 1–9 and synthetic OoD data (class 10), while fig. 19b displays distributions for all 10 classes as InD and SVHN as OoD. All results shown here use DenseNet-201 as the backbone.

## I.4 EVALUATING DENSENET ON SVHN (IND) WITH CIFAR-10 AS OOD

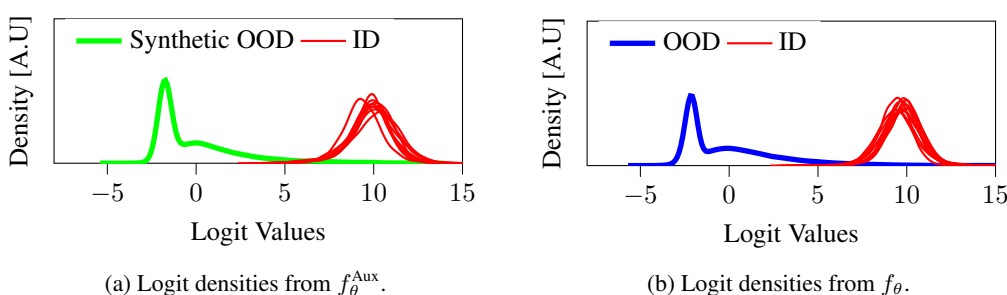

(a) Logit densities from $f_\theta^{\text{Aux}}$.

(b) Logit densities from $f_\theta$.

Figure 20: Logit density plots produced by the auxiliary classifier $f_\theta^{\text{Aux}}$ (trained on $K - 1$ CIFAR-10 classes) and the full classifier $f_\theta$ (trained on all $K$ classes). Figure 20a presents InD logit distributions for SVHN classes 1–9 and synthetic OoD data (class 10), while fig. 20b displays distributions for all 10 classes as InD and CIFAR-10 as OoD. All results shown here use DenseNet-121 as the backbone.

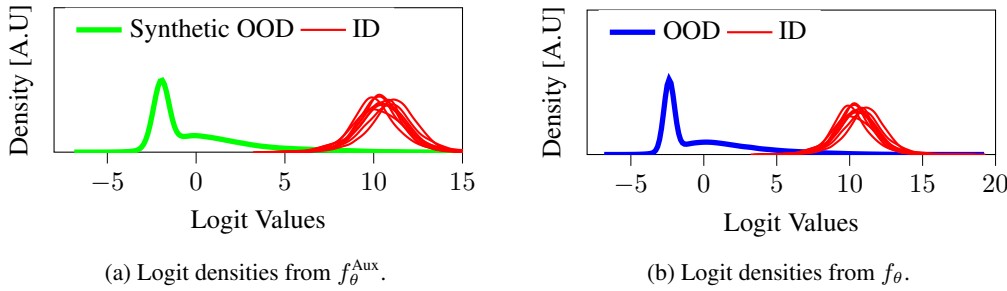

(a) Logit densities from $f_\theta^{\text{Aux}}$.

(b) Logit densities from $f_\theta$.

Figure 21: Logit density plots produced by the auxiliary classifier $f_\theta^{\text{Aux}}$ (trained on $K - 1$ CIFAR-10 classes) and the full classifier $f_\theta$ (trained on all $K$ classes). Figure 21a presents InD logit distributions for SVHN classes 1–9 and synthetic OoD data (class 10), while fig. 21b displays distributions for all 10 classes as InD and CIFAR-10 as OoD. All results shown here use DenseNet-169 as the backbone.

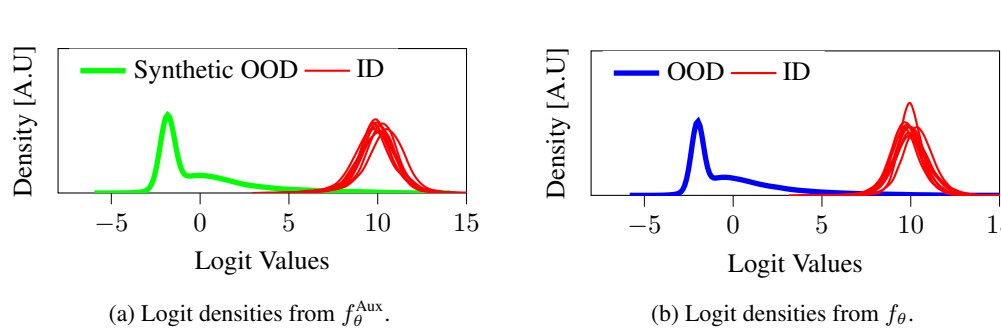

(a) Logit densities from $f_\theta^{\text{Aux}}$.

(b) Logit densities from $f_\theta$.

Figure 22: Logit density plots produced by the auxiliary classifier $f_\theta^{\text{Aux}}$ (trained on $K-1$ CIFAR-10 classes) and the full classifier $f_\theta$ (trained on all $K$ classes). Figure 22a presents InD logit distributions for SVHN classes 1–9 and synthetic OoD data (class 10), while fig. 22b displays distributions for all 10 classes as InD and CIFAR-10 as OoD. All results shown here use DenseNet-201 as the backbone.

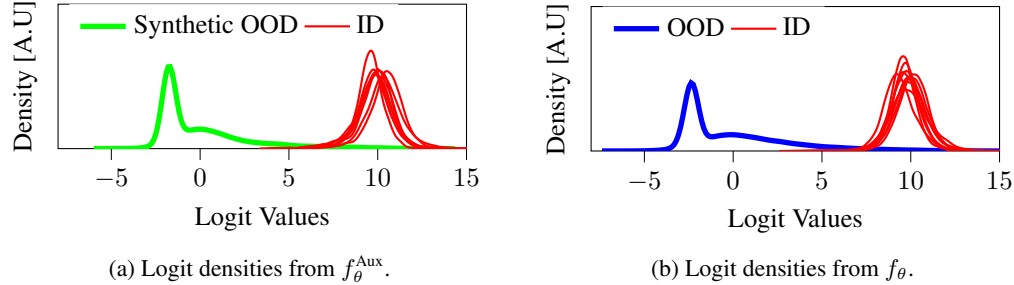

(a) Logit densities from $f_\theta^{\text{Aux}}$.

(b) Logit densities from $f_\theta$.

Figure 23: Logit density plots produced by the auxiliary classifier $f_\theta^{\text{Aux}}$ (trained on $K-1$ CIFAR-10 classes) and the full classifier $f_\theta$ (trained on all $K$ classes). Figure 23a presents InD logit distributions for SVHN classes 1–9 and synthetic OoD data (class 10), while fig. 23b displays distributions for all 10 classes as InD and CIFAR-10 as OoD. All results shown here use DenseNet-121 as the backbone.

