# OpenReview forum: "Enhancing Out-of-Distribution Detection Using Synthesized OoD Samples from In-Distribution Training Data"
_ICLR.cc/2026/Workshop/Sci4DL — Submitted to Sci4DL 2026_

### Official Review · Reviewer_7tNp · 2026-02-23

**Fit:** 3
**Significance:** 3
**Confidence:** 2

**Summary:**

The paper proposes a novel method for OOD detection in the logit space that does not require access to OOD examples. The authors build on prior work showing that in-distribution (ID) examples tend to separate along orthogonal directions, while OOD examples collapse toward the null space. Specifically, the method first trains classifiers on the target task while excluding a single class, whose embeddings then serve as artificial OOD examples for training an OOD detector. The authors support their approach with both theoretical analysis and empirical evaluation.

**Strengths:**

- The proposed method does not require labeled OOD examples and instead leverages readily available in-distribution (ID) data to approximate embeddings of realistic OOD samples.
- The authors demonstrate the efficacy of the method both theoretically and empirically, and further support their underlying assumptions with empirical evidence.

**Suggestions:**

- How does class semantic similarity affect the quality of OOD detection? Classes that are semantically close to one another may hinder the generation of high-quality OOD embeddings.
- How applicable is the method to pre-trained models, which are prevalent today? In such cases, embeddings may still appear in-distribution even for samples not seen during training

---

### Official Review · Reviewer_ypeB · 2026-02-27

**Fit:** 3
**Significance:** 2
**Confidence:** 2

**Summary:**

The authors propose an Out-of-Distribution (OOD) detection framework that synthesizes auxiliary OOD logit embeddings using entirely In-Distribution (InD) training data. By training an auxiliary model on $K-1$ classes, samples from the single excluded class collapse toward the center of the logit space, accurately mimicking OOD behavior. These synthetic OOD embeddings, combined with true InD embeddings from a model trained on all $K$ classes, are used to train a simple supervised binary classifier, establishing an explicit InD vs. OOD decision boundary

**Strengths:**

- The utilized heuristic is supported by an analytical bound to demonstrate that excluding exactly one class minimizes the approximation error compared to excluding multiple classes.
- The method effectively removes the reliance on external OOD datasets for threshold tuning and validation, which could be a significant bottleneck for traditional scoring methods.
- Despite being a short paper, it features comprehensive ablations across varying architectures (ResNets, DenseNets) and excluded classes and a competitive average performance.

**Suggestions:**

- The framework explicitly requires training the classification architecture twice (once for the auxiliary $K-1$ model, once for the full $K$ model). The authors should briefly discuss this computational cost and its scalability implications.
- The paper notes that the method relies on a balanced and sufficiently large set of InD classes. It would be helpful to briefly discuss the expected behavior on massive label spaces (e.g., ImageNet) where leaving out a single class represents a negligible perturbation, or on highly imbalanced datasets.
- The authors should briefly expand the Related Work section to differentiate their logit-level, leave-one-out approach from existing feature-space "virtual outlier synthesis" techniques (such as VOS).

---

### Meta-Review · Area_Chair_76R6 · 2026-03-01

**Recommendation:** Reject

**Metareview:**

Recommending rejection due to the lack of relevance for the workshop. This is an interesting DL engineering paper, but does not feature the scientific method.

---

### Decision · Program_Chairs · 2026-03-02

Reject